# Multimodal Analogical Reasoning over Knowledge Graphs

**Ningyu Zhang**[1][*]  **Lei Li**[1][*]  **Xiang Chen**[1][*]  **Xiaozhuan Liang**[1]    **Shumin Deng**[2]    **Huajun Chen**[1][†]

[1]Zhejiang University, AZFT Joint Lab for Knowledge Engine
[2]National University of Singapore
{zhangningyu,leili21,xiang_chen,liangxiaozhuan,231sm,huajunsir}@zju.edu.cn

## Abstract

Analogical reasoning is fundamental to human cognition and holds an important place in various fields. However, previous studies mainly focus on single-modal analogical reasoning and ignore taking advantage of structure knowledge. Notably, the research in cognitive psychology has demonstrated that information from multimodal sources always brings more powerful cognitive transfer than single modality sources. To this end, we introduce the new task of multimodal analogical reasoning over knowledge graphs, which requires multimodal reasoning ability with the help of background knowledge. Specifically, we construct a **M**ultimodal **A**nalogical **R**easoning data**S**et (**MARS**) and a multimodal knowledge graph **MarKG**. We evaluate with multimodal knowledge graph embedding and pre-trained Transformer baselines, illustrating the potential challenges of the proposed task. We further propose a novel model-agnostic **M**ultimodal **a**nalogical **r**easoning framework with **T**ransformer (**MarT**) motivated by the structure mapping theory, which can obtain better performance. We hope our work can deliver benefits and inspire future research[1].

## 1 Introduction

Analogical reasoning – the ability to perceive and use relational similarity between two situations or events – holds an important place in human cognition (Johnson-Laird, 2006; Wu et al., 2020; Bengio et al., 2021; Chen et al., 2022a) and can provide back-end support for various fields such as education (Thagard, 1992), creativity (Goel, 1997), thus appealing to the AI community. Early, Mikolov et al. (2013b); Gladkova et al. (2016a); Ethayarajh et al. (2019a) propose visual analogical reasoning aiming at lifting machine intelligence in Computer Vision (CV) by associating vision with relational, structural, and analogical reasoning. Meanwhile, researchers of Natural Language Processing (NLP) hold the connectionist assumption (Gentner, 1983) of linear analogy (Ethayarajh et al., 2019b); for example, the relation between two words can be inferred through vector arithmetic of word embeddings. However, it is still an open question whether artificial neural networks are also capable of recognizing analogies among different modalities.

Note that humans can quickly acquire new abilities based on finding a common relational system between two exemplars, situations, or domains. Based on Mayer's Cognitive Theory of multimedia learning (Hegarty & Just, 1993; Mayer, 2002), human learners often perform better on tests with analogy when they have learned from multimodal sources than single-modal sources. Evolving from recognizing single-modal analogies to exploring multimodal reasoning for neural models, we emphasize the importance of a new kind of analogical reasoning task with Knowledge Graphs (KGs).

In this paper, we introduce the task of multimodal analogical reasoning over knowledge graphs to fill this blank. Unlike the previous multiple-choice QA setting, we directly predict the analogical target and formulate the task as **link prediction without explicitly providing relations**. Specifically, the task can be formalized as $(e_h, e_t) : (e_q, ?)$ with the help of background multimodal knowledge graph

---

[*]Equal contribution and shared co-first authorship.
[†]Corresponding author.
[1]Code and datasets are available in `https://github.com/zjunlp/MKG_Analogy`.

$\mathcal{G}$, in which $e_h$, $e_t$ or $e_q$ have different modalities. We collect a **M**ultimodal **A**nalogical **R**easoning data**S**et (**MARS**) and a multimodal knowledge graph **MarKG** to support this task. These data are collected and annotated from seed entities and relations in E-KAR (Chen et al., 2022a) and BATs (Gladkova et al., 2016a), with linked external entities in Wikidata and images from Laion-5B (Schuhmann et al., 2021).

To evaluate the multimodal analogical reasoning process, we follow the guidelines from psychological theories and conduct comprehensive experiments on MARS with multimodal knowledge graph embedding baselines and multimodal pre-trained Transformer baselines. We further propose a novel **M**ultimodal **a**nalogical **r**easoning framework with **T**ransformer, namely **MarT**, which is readily pluggable into any multimodal pre-trained Transformer models and can yield better performance.

To summarize, our contributions are three-fold: (1) We advance the traditional setting of analogy learning by introducing a new multimodal analogical reasoning task. Our work may open up new avenues for improving analogical reasoning through multimodal resources. (2) We collect and build a dataset MARS with a multimodal knowledge graph MarKG, which can be served as a scaffold for investigating the multimodal analogy reasoning ability of neural networks. (3) We report the performance of various multimodal knowledge graph embedding, multimodal pre-trained Transformer baselines, and our proposed framework MarT. We further discuss the potential of this task and hope it facilitates future research on zero-shot learning and domain generalization in both CV and NLP.

## 2 BACKGROUND

### 2.1 ANALOGICAL REASONING IN PSYCHOLOGICAL

To better understand analogical reasoning, we introduce some crucial theories from cognitive psychology, which we take as guidelines for designing the multimodal analogical reasoning task.

**Structure Mapping Theory (SMT)** (Gentner, 1983). SMT is a theory that takes a fundamental position in analogical reasoning. Specifically, SMT emphasizes that humans conduct analogical reasoning depending on the shared *relations* structure rather than the superficial *attributes* of domains and distinguishes analogical reasoning with literal similarity. Minnameier (2010) further develops the inferential process of analogy into three steps: abduction, mapping and induction, which inspires us to design benchmark baselines for multimodal analogical reasoning.

**Mayer's Cognitive Theory** (Hegarty & Just, 1993; Mayer, 2002). Humans live in a multi-source heterogeneous world and spontaneously engage in analogical reasoning to make sense of unfamiliar situations in everyday life (Vamvakoussi, 2019). Mayer's Cognitive Theory shows that human learners often perform better on tests of recall and transfer when they have learned from multimodal sources than single-modal sources. However, relatively little attention has been paid to multimodal analogical reasoning, and it is still unknown whether neural network models have the ability of multimodal analogical reasoning.

### 2.2 ANALOGICAL REASONING IN CV AND NLP

**Visual Analogical Reasoning.** Analogical reasoning in CV aims at lifting machine intelligence by associating vision with relational, structural, and analogical reasoning (Johnson et al., 2017; Prade & Richard, 2021; Hu et al., 2021; Malkinski & Mandziuk, 2022). Some datasets built in the context of Raven's Progressive Matrices (RPM) are constructed, including PGM (Santoro et al., 2018) and RAVEN (Zhang et al., 2019). Meanwhile, Hill et al. (2019) demonstrates that incorporating structural differences with structure mapping in analogical visual reasoning benefits the machine learning models. Hayes & Kanan (2021) investigates online continual analogical reasoning and demonstrates the importance of the selective replay strategy. However, these aforementioned works still focus on analogy reasoning among visual objects while ignoring the role of complex texts.

**Natural Language Analogical Reasoning.** In the NLP area, early attempts devote to word analogy recognition (Mikolov et al., 2013b; Gladkova et al., 2016a; Jurgens et al., 2012; Ethayarajh et al., 2019a; Gladkova et al., 2016b) which can often be effectively solved by vector arithmetic for neural word embeddings Word2Vec (Mikolov et al., 2013a) and Glove (Pennington et al., 2014). Recent studies have also evaluated on the pre-trained language models (Devlin et al., 2019; Brown et al.,

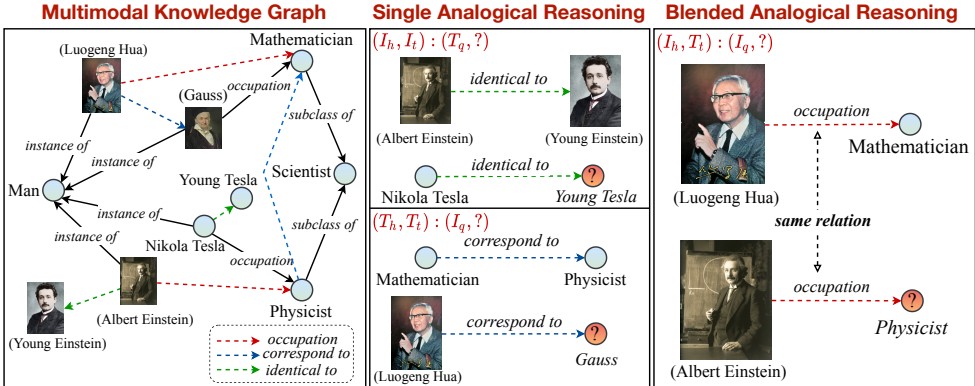

Figure 1: Overview of the Multimodal Analogical Reasoning task. We divide the task into single and blended settings with a multimodal knowledge graph. Note that the relation marked by dashed arrows (--→) and the text around parentheses under images are **only for annotation** and **not provided in the input**.

2020; Ushio et al., 2021). However, word analogies mainly measure the quality of word representations and do not explore the analogical reasoning ability of models. Thus, Chen et al. (2022a) builds a knowledge-intensive benchmark to evaluate the analogical reasoning ability of neural models. Nevertheless, Chen et al. (2022a) mainly focuses on reasoning in the textual domain and does not consider using external knowledge graphs. In this work, we take the first step to investigate multimodal analogical reasoning over knowledge graphs.

# 3 THE MULTIMODAL ANALOGICAL REASONING TASK

## 3.1 TASK DEFINITION

In this section, we introduce the task of Multimodal Analogical Reasoning that can be formulated as link prediction **without explicitly providing relations**. As shown in Figure 1, given an analogy example $(e_h, e_t)$ and a question-answer entity pair $(e_q, ?)$ where $e_h, e_t, e_q \in \mathcal{E}_a$ and $\mathcal{E}_a \in \mathcal{E}$, the goal of analogical reasoning is to predict the missing entity $e_a \in \mathcal{E}_a$. Moreover, multimodal analogical reasoning is based on background multimodal knowledge graph $\mathcal{G} = (\mathcal{E}, \mathcal{R}, \mathcal{I}, \mathcal{T})$, where $\mathcal{E}$ and $\mathcal{R}$ are sets of entities and relations, $\mathcal{I}$ and $\mathcal{T}$ represent images and textual descriptions of entities. Note that the relations of $(e_h, e_t)$ and $(e_q, e_a)$ are identical but unavailable, and the relation structure can be analogized implicitly from source domain to target domain without knowing the relations. Specifically, the task can be formalized as $(e_h, e_t) : (e_q, ?)$, further divided into *Single Analogical Reasoning* and *Blended Analogical Reasoning* according to different modalities of $e_h, e_t, e_q$ and $e_a$.

**Single Analogical Reasoning.** In this setting, the analogy example and the question-answer entity pair involve only one modality. As shown in the middle column of Figure 1, the modalities of the analogy example $(e_h, e_t)$ are identical and opposite to the analogy question-answer pair $(e_q, e_a)$. Based on both visual and textual modalities, this setting can be further divided into $(I_h, I_t) : (T_q, ?)$ and $(T_h, T_t) : (I_q, ?)$ where $I_h, T_h$ represent the modality of $e_h$ is visual or textual respectively.

**Blended Analogical Reasoning.** In the setting, the modality of analogy example $(e_h, e_t)$ are unidentical, which is similar to real-world human cognition and perception[2]. Note that Mayer's theory indicates that humans can have powerful transfer and knowledge recall abilities in multimodal scenarios. Inspired by this, we propose the blended analogical reasoning that can be formalized as $(I_h, T_t) : (I_q, ?)$, which means the modalities between $e_h$ ($e_q$) and $e_t$ ($e_a$) are different.

## 3.2 DATA COLLECTION AND PREPROCESSING

We briefly introduce the construction process of the dataset in Figure 2. Firstly, we collect a multimodal knowledge graph dataset MarKG and a multimodal analogical reasoning dataset MARS,

---

[2]For example, humans invented hieroglyphics by analogy from the concrete world.

Figure 2: An illustration of data collection and processing steps to create MARS and MarKG.

| Dataset | Size (train / dev / test) | KB | Modality | # Entity | # Relation | # Images | Knowledge Intensive | Task Format |
|---|---|---|---|---|---|---|---|---|
| RAVEN | 42,000 / 14,000 / 14,000 | ✗ | Vision | - | 8 | 1,120,000 | ✗ | Classification |
| SAT | 0 / 37 / 337 | ✗ | Text | - | 19 | - | ✗ | Linear Word Analogy |
| Google | 0 / 50 / 500 | ✗ | Text | 919 | 14 | - | ✗ | Linear Word Analogy |
| BATs | 0 / 199 / 1,799 | ✗ | Text | 6,218 | 40 | - | ✗ | Linear Word Analogy |
| E-KAR | 870 / 119 / 262 | ✗ | Text | 2,032 | 28 | - | ✔ | Multiple Choice QA |
| MARS | 10,685 / 1,228 / 1,415 | MarKG | Vision+Text | 2,063 | 27 | 13,398 | ✔ | Entity Prediction |

Table 1: Comparison between MARS and previous analogical reasoning datasets. "KB" refers to the knowledge base, # denotes the number. "Knowledge Intensive" means reasoning requires external knowledge. Our MarKG focuses on knowledge-intensive reasoning across multiple modalities.

which are developed from seed entities and relations in E-KAR (Chen et al., 2022a) and BATs (Gladkova et al., 2016a). Secondly, we link these seed entities into the free and open knowledge base Wikidata[3] for formalization and normalization. Thirdly, to acquire the image data, we further search from the Google engine and query from the multimodal data Laion-5B (Schuhmann et al., 2021) by the text descriptions of entities. Then, an image validation strategy is applied to filter low-quality images. Lastly, we sample high-quality analogy data to construct MARS. A detailed description of the data collection and processing to create our datasets are in Appendix B.1 and B.2.

## 3.3 DATASET STATISTICS

MARS is the evaluation dataset of the multimodal analogical reasoning task that contains analogy instances, while MarKG can provide the relative structure information of those analogy entities retrieved from Wikidata. The statistics of MARS and MarKG are shown in Table 1 and Table 5. MarKG has 11,292 entities, 192 relations and 76,424 images, include 2,063 analogy entities and 27 analogy relations. MARS has 10,685 training, 1,228 validation and 1,415 test instances, which are more significant than previous language analogy datasets. The original intention of MarKG is to provide prior knowledge of analogy entities and relations for better reasoning. We release the dataset with a leaderboard at https://zjunlp.github.io/project/MKG_Analogy/. More details including **quality control** can be found in Appendix B.3.

## 3.4 EVALUATION METRICS

Previous study (Chen et al., 2022a) adopts the multiple-choice QA to conduct analogical reasoning and leverage the accuracy metric for evaluation. However, the multiple-choice QA setting may struggle to handle the *one-to-more entities*, which is very common in real-world analogy scenarios. Thus, we formulate the task as link prediction that directly predicts the answer entity $e_a \in \mathcal{E}_a$. Our evaluation metrics include Hits@k scores (proportion of valid entities ranked in top k) and MRR (reciprocal value of the mean rank of correct entities). More details can be found in Appendix B.4.

## 4 BENCHMARK METHODS

In this section, we introduce some baselines to establish the initial benchmark results on MARS, including multimodal knowledge graph embedding baselines and multimodal pre-trained Transformer baselines. We further propose MarT: a multimodal analogical reasoning framework with Trans-

---

[3]https://www.wikidata.org

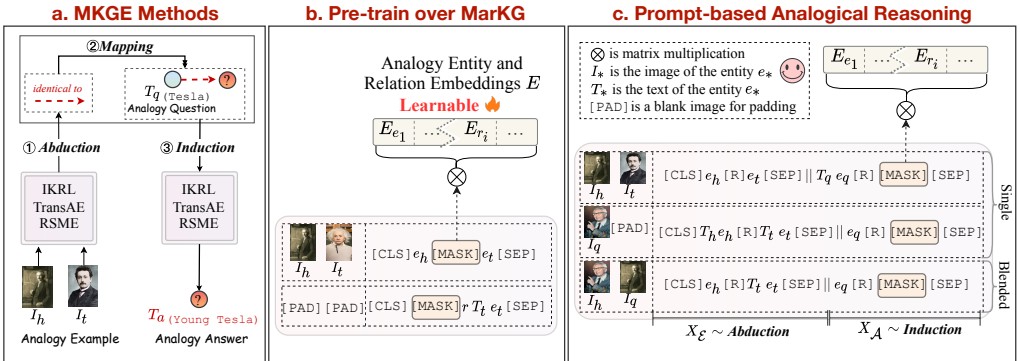

Figure 3: Overview of baseline methods. (a) Pipeline of MKGE methods for multimodal analogical reasoning. (b) and (c) are two stages of multimodal pre-trained Transformer (MPT) baselines.

former, which can capture fine-grained associations between one analogy example and one analogy question-answer pair for better multimodal analogy abilities.

## 4.1 MULTIMODAL KNOWLEDGE GRAPH EMBEDDING BASELINES

We consider three multimodal knowledge embedding (MKGE) approaches as our baselines, including **IKRL** (Xie et al., 2017), **TransAE** (Wang et al., 2019), and **RSME** (Wang et al., 2021). These methods are typically based on TransE (Bordes et al., 2013) or ComplEx (Trouillon et al., 2016) and combine with visual encoders to encode images for multimodal knowledge representation learning. They can not be directly applied to the multimodal analogical reasoning task. To successfully utilize MKGE methods, we first pre-train them on MarKG to obtain entity embeddings and then follow the structure-mapping theory (Minnameier, 2010) to leverage the *Abduction-Mapping-Induction* as **explicit pipline** steps for MKGE methods. As shown in Figure 3.a, *Abduction* aims to predict the relation $r$ of $(e_h, e_t)$ similar to the relation classification task, *Mapping* represents that the structural relation is mapped onto entity candidates, analogous to template-filling, and *Induction* utilizes the relation $r$ to predict the tail entity of $(e_q, r, ?)$ similar to the link prediction task.

Despite the previous MKGE methods achieving excellent performance for KG-related tasks, the backbone, such as TransE, is not designed for analogy reasoning, which may hinder performance. Thus, we replace the backbone of MKGE methods with ANALOGY Liu et al. (2017) that models analogical structure explicitly as baselines.

## 4.2 MULTIMODAL PRE-TRAINED TRANSFORMER BASELINES

We select multimodal pre-trained Transformer (MPT) approaches including the single-stream models **VisualBERT** (Li et al., 2019), **ViLT** (Kim et al., 2021), the dual-stream model **ViLBERT** (Lu et al., 2019), and the mixed-stream model **FLAVA** (Singh et al., 2022) and **MKGformer** Chen et al. (2022b) as the strong baselines. However, the current multimodal pre-trained Transformer cannot directly deal with analogical reasoning. To address the bottleneck above, we devise an **end-to-end** approach to empower the MPT with analogical reasoning ability. As shown in Figure 3, we first leverage MarKG to pre-train the model over sparse MarKG to obtain the representation of entities and relations. We then present the prompt-based analogical reasoning over MARS.

### 4.2.1 PRE-TRAIN OVER MARKG

We represent the entities $e \in \mathcal{E}$ and relations $r \in \mathcal{R}$ as special tokens and denote $E$ as the learnable embedding of these special tokens in the word vocabulary of language models. In the pre-train stage, we design masked entity and relation prediction like the Masked Language Modeling (MLM) task to learn the embeddings of the special tokens over the MarKG dataset. As shown in Figure 3.b, we devise a prompt template to convert the input as predicting the missing entity and relation via [MASK] token. In addition, we mix missing relation and entity prediction in the pre-train stage and consider different modalities of input entities. Specifically, we represent the visual entity $e_h$ by

its image $I_h$ and special entity embedding $E_{e_h}$, and the text entity $e_t$ by its text description $T_t$ and special entity embedding $E_{e_t}$, respectively. Benefiting from the mixed entity and relation prediction with the multimodal entity in the pre-train stage, we can obtain KG embedding with multimodal semantics over the current knowledge graph MarKG.

### 4.2.2 PROMPT-BASED ANALOGICAL REASONING

Based on the above-pre-trained entity and relation embeddings over MarKG, we propose prompt-based analogical reasoning with implicit structure mapping on downstream MARS.

Taking the blended analogical reasoning as an example, we feed the analogy example $(I_h, T_t)$ and analogy question-answer pair $(I_q, ?)$ as input, and the goal is to predict the missing answer entity $e_a \in \mathcal{E}_a$. We leverage an analogical prompt template to convert the input as follows:

$$\mathcal{T}_{(I_h, T_t, I_q)} = \mathcal{T}_{\mathcal{E}} \parallel \mathcal{T}_{\mathcal{A}} = I_h \, I_q \, \texttt{[CLS]} \, e_h \, \texttt{[R]} \, T_t \, e_t \, \texttt{[SEP]} \parallel e_q \, \texttt{[R]} \, \texttt{[MASK]} \, \texttt{[SEP]}, \quad (1)$$

where $\parallel$ represents concatenate operation in the template input, $I_h$ and $I_q$ represent the images of the entity $e_h$ and $e_q$, $T_t$ is the text description of the entity $e_t$. Moreover, $e_h, e_t, e_q$ are entity ids and will be encoded to special entity tokens $E_{e_h}, E_{e_t}, E_{e_q}$ in word embedding layer. Since the relations are not explicitly provided in the actual analogical reasoning task, we assign $\texttt{[R]}$ as a special token to denote the explicit relation between $(I_h, T_t)$, which is initialized with the average relation embeddings. Finally, we train the model to predict the $\texttt{[MASK]}$ over the special token embedding $E$ via cross-entropy loss, which likes the MLM task.

**Remark 1** *We summarize the two parts of $\mathcal{T}_{\mathcal{E}}$ and $\mathcal{T}_{\mathcal{A}}$ in the template as the implicit Abduction and Induction respectively, which are unified in an end-to-end learning manner with prompt tuning. In addition, the analogical reasoning is reformulated as predicting the $\texttt{[MASK]}$ over the multimodal analogy entity embeddings to obtain $e_a$.*

### 4.3 MART: A MULTIMODAL ANALOGICAL REASONING FRAMEWORK WITH TRANSFORMER

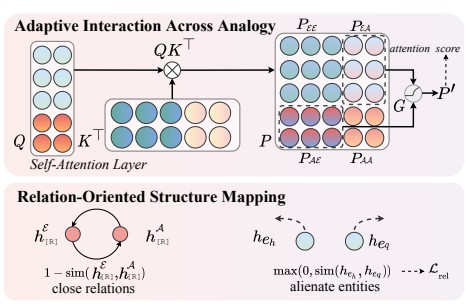

Figure 4: The MarT framework.

Although the approach above-mentioned can enable multimodal pre-trained Transformer models to multimodal analogical reasoning, they only superficially consider implicit *Abduction* and *Induction*, ignoring the fine-grained associations between the analogy example and analogy question-answer pair.

**Adaptive Interaction Across Analogy.** Since the analogy question may interfere with the representation of the analogy example and the inevitable noisy data issue, we propose adaptive interaction across analogy in encoding process to interact between the analogy example and question-answer pair adaptively, as shown in Figure 4. Denote the input to a Transformer layer as $X = [X_{\mathcal{E}} \parallel X_{\mathcal{A}}]$, where $X_{\mathcal{E}}$ and $X_{\mathcal{A}}$ denote the hidden representation of analogy example $\mathcal{T}_{\mathcal{E}}$ and question-answer pair $\mathcal{T}_{\mathcal{A}}$ respectively. In each attention head of layer, the query and key representation can be formalized as:

$$Q = XW^Q = \begin{pmatrix} X_{\mathcal{E}} \\ X_{\mathcal{A}} \end{pmatrix} W^Q = \begin{pmatrix} Q_{\mathcal{E}} \\ Q_{\mathcal{A}} \end{pmatrix}, K = XW^K = \begin{pmatrix} X_{\mathcal{E}} \\ X_{\mathcal{A}} \end{pmatrix} W^K = \begin{pmatrix} K_{\mathcal{E}} \\ K_{\mathcal{A}} \end{pmatrix}, \quad (2)$$

where $W^Q, W^K$ are project matrices. A similar expression also holds for values $V$. Then the attention probability matrix $P$ can be defined in terms of four sub-matrices:

$$P = QK^{\top} = \begin{pmatrix} Q_{\mathcal{E}} \\ Q_{\mathcal{A}} \end{pmatrix} (K_{\mathcal{E}}^{\top}, K_{\mathcal{A}}^{\top}) = \begin{pmatrix} Q_{\mathcal{E}}K_{\mathcal{E}}^{\top} & Q_{\mathcal{E}}K_{\mathcal{A}}^{\top} \\ Q_{\mathcal{A}}K_{\mathcal{E}}^{\top} & Q_{\mathcal{A}}K_{\mathcal{A}}^{\top} \end{pmatrix} = \begin{pmatrix} P_{\mathcal{E}\mathcal{E}} & P_{\mathcal{E}\mathcal{A}} \\ P_{\mathcal{A}\mathcal{E}} & P_{\mathcal{A}\mathcal{A}} \end{pmatrix} \quad (3)$$

where $P_{\mathcal{E}\mathcal{E}}, P_{\mathcal{A}\mathcal{A}}$ (diagonal of $P$) are intra-analogy attentions and $P_{\mathcal{E}\mathcal{A}}, P_{\mathcal{A}\mathcal{E}}$ (anti-diagonal of $P$) are inter-analogy attentions. We use the gate $G$ to regulate the inter-analogy interactions adaptively:

$$P' = G \odot P = \begin{pmatrix} 1 & g_{\mathcal{E}\mathcal{A}} \\ g_{\mathcal{A}\mathcal{E}} & 1 \end{pmatrix} \odot \begin{pmatrix} P_{\mathcal{E}\mathcal{E}} & P_{\mathcal{E}\mathcal{A}} \\ P_{\mathcal{A}\mathcal{E}} & P_{\mathcal{A}\mathcal{A}} \end{pmatrix} = \begin{pmatrix} P_{\mathcal{E}\mathcal{E}} & g_{\mathcal{E}\mathcal{A}}P_{\mathcal{E}\mathcal{A}} \\ g_{\mathcal{A}\mathcal{E}}P_{\mathcal{A}\mathcal{E}} & P_{\mathcal{A}\mathcal{A}} \end{pmatrix} \quad (4)$$

where $G \in \mathbb{R}^{2\times 2}$ is adaptive association gate which has two learnable variables $g_{\mathcal{E}\mathcal{A}}, g_{\mathcal{A}\mathcal{E}} \in [0, 1]$.

| Method | Baselines | Backbone | Hits@1 | Hits@3 | Hits@5 | Hits@10 | MRR |
|---|---|---|---|---|---|---|---|
| MKGE | IKRL | TransE | 0.254 | 0.285 | 0.290 | 0.304 | 0.274 |
|  | TransAE | TransE | 0.203 | 0.233 | 0.241 | 0.253 | 0.223 |
|  | RSME | ComplEx | 0.255 | 0.274 | 0.282 | 0.291 | 0.268 |
|  | IKRL | ANALOGY | 0.266 | 0.294 | 0.301 | 0.310 | 0.283 |
|  | TransAE | ANALOGY | 0.261 | 0.285 | 0.289 | 0.293 | 0.276 |
|  | RSME | ANALOGY | 0.266 | 0.298 | 0.307 | 0.311 | 0.285 |
| MPT | VisualBERT | Single-Stream | 0.247 | 0.281 | 0.289 | 0.303 | 0.269 |
|  | ViLT | Single-Stream | 0.235 | 0.266 | 0.274 | 0.286 | 0.257 |
|  | ViLBERT | Dual-Stream | 0.252 | 0.308 | 0.320 | 0.338 | 0.287 |
|  | FLAVA | Mixed-Stream | 0.257 | 0.299 | 0.312 | 0.325 | 0.284 |
|  | MKGformer | Mixed-Stream | 0.293 | 0.335 | 0.344 | 0.367 | 0.321 |
|  | MarT_VisualBERT | Single-Stream | 0.261 | 0.292 | 0.308 | 0.321 | 0.284 |
|  | MarT_ViLT | Single-Stream | 0.245 | 0.275 | 0.287 | 0.303 | 0.266 |
|  | MarT_ViLBERT | Dual-Stream | 0.256 | 0.312 | 0.327 | 0.347 | 0.292 |
|  | MarT_FLAVA | Mixed-Stream | 0.264 | 0.303 | 0.309 | 0.319 | 0.288 |
|  | MarT_MKGformer | Mixed-Stream | **0.301** | **0.367** | **0.380** | **0.408** | **0.341** |

Table 2: The main performance results on MARS. We report pipeline baselines with multimodal knowledge graph embedding (MKGE) methods and replace their backbone models with analogy-aware model ANALOGY. We also utilize our MarT on end-to-end baselines with multimodal pre-trained Transformer (MPT) methods and obtain the best performance in MarT_MKGformer.

**Remark 2** *On the one hand, the query from $\mathcal{T}_\mathcal{A}$ may interfere with the example from $\mathcal{T}_\mathcal{E}$. On the other hand, $\mathcal{T}_\mathcal{E}$ may have a weaker impact on $\mathcal{T}_\mathcal{A}$ in noisy data. Adaptive association gates can increase and decrease inter-analogy interaction automatically based on the intimacy of $\mathcal{T}_\mathcal{E}$ and $\mathcal{T}_\mathcal{A}$.*

**Relation-Oriented Structure Mapping.** The structure mapping theory emphasizes the relation transfer rather than object similarity in analogical reasoning, it is *relations between objects, rather than attributes of objects, are mopped from base to target*. For example, *battery* can make an analogy to *reservoir* because they both store potential, rather than their shapes being cylindrical. Motivated by this, we propose the **relaxation loss** to bring the relations closer and alienate the entities:

$$\mathcal{L}_{\text{rel}} = \frac{1}{|\mathcal{S}|} \sum_i^{|\mathcal{S}|} (\underbrace{1 - \text{sim}(h_{[\text{R}]}^\mathcal{E}, h_{[\text{R}]}^\mathcal{A})}_{\text{close relations}} + \underbrace{\max(0, \text{sim}(h_{e_h}, h_{e_q}))}_{\text{alienate entities}}) \tag{5}$$

where $|\mathcal{S}|$ is the total number of the training set $\mathcal{S}$, $h_{[\text{R}]}^\mathcal{E}$ is the hidden feature of $[\text{R}]$ in analogy example $\mathcal{T}_\mathcal{E}$ output from the MLM head, $\text{sim}(\cdot)$ is the cosine similarity. We leverage the masked entity prediction task to obtain the answer entity $e_a$ with a cross-entropy loss:

$$\mathcal{L}_{\text{mem}} = -\frac{1}{|\mathcal{S}|} \sum_{(e_h, e_t, e_q, e_a) \in \mathcal{S}} \log(p([\text{MASK}] = e_a) | \mathcal{T}_{(e_h, e_t, e_q)}) \tag{6}$$

Afterwards, we interpolate the relaxation loss $\mathcal{L}_{\text{rel}}$ and the masked entity prediction loss $\mathcal{L}_{\text{mem}}$ using parameter $\lambda$ to produce the final loss $\mathcal{L}$:

$$\mathcal{L} = \lambda \mathcal{L}_{\text{rel}} + (1 - \lambda) \mathcal{L}_{\text{mem}} \tag{7}$$

**Remark 3** *The relaxation loss is composed of pull-in and pull-away that correspond to the close relation and alienate entity terms, respectively, which can constrain the model's focus on relation structure transfer and implicitly realize the Structure Mapping process.*

## 5 RESULTS AND ANALYSIS

### 5.1 MAIN RESULTS

The main performance results of all benchmark methods can be seen in Table 2. In general, we find the performance of multimodal knowledge graph embedding (MKGE) baselines and multimodal pre-trained Transformer (MPT) baselines is comparable except MKGformer, which establishes a

| Model | Hits@1 | Hits@3 | Hits@5 | Hits@10 | MRR |
|---|---|---|---|---|---|
| TransAE | 0.203 | 0.233 | 0.241 | 0.253 | 0.223 |
| w/o MarKG | 0.191 | 0.224 | 0.235 | 0.245 | 0.214 |
| MarT_ViLBERT | 0.256 | 0.312 | 0.327 | 0.347 | 0.292 |
| w/o MarKG | 0.253 | 0.292 | 0.297 | 0.310 | 0.270 |
| w/o Analogy example | 0.113 | 0.143 | 0.162 | 0.179 | 0.138 |
| MarT_MKGformer | **0.301** | **0.367** | **0.380** | **0.408** | **0.341** |
| w/o MarKG | 0.270 | 0.305 | 0.309 | 0.315 | 0.289 |
| w/o Relaxation loss | 0.295 | 0.349 | 0.373 | 0.399 | 0.332 |
| w/o Adaptive interaction | 0.285 | 0.345 | 0.365 | 0.395 | 0.324 |
| w/o MarT | 0.293 | 0.335 | 0.344 | 0.367 | 0.321 |
| w/o Analogy example | 0.101 | 0.123 | 0.132 | 0.149 | 0.120 |

Table 4: Ablation experiments on MARS. w/o MarKG refers to the model without pre-training on MarKG dataset. w/o MarT refers to ablate all components of MarT that equivalents to MKGformer.

competitive baseline of MARS. In addition, when replacing the backbone of MKGE methods with ANALOGY that models analogical structure explicitly, the performance is significantly improved. Meanwhile, the MPT models without analogy-related structures obtain substantial performance with the analogical reasoning ability enhanced by MarT. For example, although MKGformer achieves outstanding performance, MarT_MKGformer further improves and obtains state-of-the-art performance, exceeding other methods by 4.9%-12.4% points in the MRR metric. It reveals that the MarT framework stimulates the ability of the Transformer-based model for multimodal analogical reasoning. We also report the pre-training results on MarKG in Appendix C.2.

## 5.2 GENERALIZE TO NOVEL RELATION

Making analogies from one domain to another novel domain is a fundamental ingredient for human creativity. In this section, we conduct a novel relation transfer experiment (including both task settings) to measure how well the models generalize by analogy to unfamiliar relations. Specifically, we randomly split the 27 analogy relations into the source and target relations. The models are then trained on the source and tested on the novel target relations. As shown in Table 3, we observe that

| Model | Novel Relation Transfer | | | |
|---|---|---|---|---|
| | Hits@1 | Hits@3 | Hits@10 | MRR |
| MarT_MKGformer | 0.254 | 0.285 | 0.292 | 0.273 |
| w/o MarKG | 0.217 | 0.228 | 0.231 | 0.224 |
| w/ Full MARS | 0.365 | 0.419 | 0.433 | 0.395 |

Table 3: Results of MKGformer on novel relation generalization. "w/ Full MARS" is the result trained with full data (upper bound).

MarT_MKGformer can indeed learn to make sense of unfamiliar relations, respectively. We further evaluate the model without pre-training on MarKG and find the performance decreased, which indicates that the structure knowledge provided by MarKG is critical for generalization. Note that the novel relation transfer setting is somewhat similar to zero-shot or domain generalization, and we hope our work can benefit other communities.

## 5.3 ABLATION STUDY

To validate the effectiveness of MarKG and MarT, we conduct an ablation study as shown in Table 4. We observe that discarding pre-train on MarKG results in worse performance for both MKGE and MPT baselines. It indicates that the knowledge structure information provided by MarKG helps learn the representation of entities and relations, further benefiting analogical reasoning. We also find that the performance clearly drops when ablating each component of MarT and reaches the valley when ablating all, proving the effectiveness of each analogical component of our MarT. Moreover, we ablate the analogy example in the input and find the performance drops a lot, which reveals the importance of analogical prompts.

## 5.4 ANALYSIS

**Analysis across Different Sub-Tasks.** In previous Table 2, we are amazed by ANALOGY significantly improving the performance of MKGE baselines. Therefore, we further compare the perfor-

| Task Setting | Analogical Example | Question-Answer Pair | Top-3 Entity | | | | Gold Rank |
|---|---|---|---|---|---|---|---|
| $(I_h, I_t)$ ↓ $(T_q, ?)$ | Qinghai Lake → *correspond to* → Inland Lake | campaign → *correspond to* → battle | MKGformer | *film* | increment | scheme | 207 |
| | | | MKGformer* | ***battle*** | war | siege | 1 |
| | | | TransAE | *life* | aircraft | ocean | 254 |
| | | | TransAE* | ***battle*** | reaction | court | 1 |
| $(I_h, T_t)$ ↓ $(I_q, ?)$ | Panax notoginseng → *instance of* → Traditional Chinese Medicine | Apple → *instance of* → fruit | MKGformer | *bread* | capital | phone | 80 |
| | | | MKGformer* | ***fruit*** | dried fruit | citrus | 1 |
| | | | TransAE | *Citrus* | grain | shipping | 125 |
| | | | TransAE* | *plant* | Citrus | dessert | 6 |

Figure 6: Case examples of MARS. We show the analogy example and analogy question-answer pair with their implicit relations. "Top-3 Entity" means top-3 ranking entities in the prediction. "Gold Rank" refers to the rank of the gold answer entity in the prediction. * denotes the baseline model with analogical components (MarT or ANALOGY).

mance of vanilla baselines to the addition of analogical components in different sub-task settings. As shown in Figure 5, we observe that vanilla TransAE performs poorly in the blended task setting. However, when replacing the backbone TransE with ANALOGY, TransAE is competent in blended analogical reasoning setting and even outperforms the single setting. On the other side, RSME with ComplEx as backbone can handle the blended setting reluctantly but perform worse than the single setting. ANALOGY improves the performance of RSME in this situation. Meanwhile, MarT further explores the potential of MKGformer and improves its performance in various tasks. All in all, the analogical components consistently improve the multimodal analogical reasoning ability of all baseline methods, especially in blended analogical reasoning, which supports Mayer's theory (Mayer, 2002) that **analogical reasoning is more affinity for multimodal scenarios**.

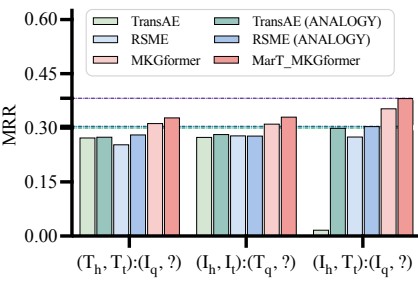

Figure 5: Performance on MARS in different sub-task settings.

**Case Analysis.** As shown in Figure 6, we provide case analysis and observe that the top ranking entities (film, life, etc.) of the baselines without analogical components are usually irrelevant to the question entity "campaign"[4]. Analogical components make the predictions more reasonable and successfully predict the answer entity "battle". In the difficult blended analogical reasoning setting, the blended modal input of visual and text is challenging. We find that vanilla MKGformer and TransAE fail to understand the visual semantic of "apple" and incorrectly linked with "capital, phone, shipping" that related to "Apple Company". We also notice that TransAE with ANALOGY as backbone significantly decreases the prediction error but incorrectly predicts "plant" as the top-1 entity due to the interference of "Panax notoginseng". On the contrary, MarT_MKGformer with relaxation loss can alienate the entities and focus on relation structures transfer and obtain reasonable predictions. These observations reveal that multimodal analogical reasoning is a highly challenging task, and analogy-aware components could enhance the analogical ability of models. Besides, we discuss **limitations** in Appendix A and provide a comprehensive error analysis in Appendix D.

## 6 DISCUSSION AND CONCLUSION

In this work, we introduce the new task of multimodal analogical reasoning over knowledge graphs.Preliminary experiments show that this task brings a rather difficult challenge and is worth further exploration. Besides evaluating the analogical reasoning ability of models, there are some potential applications to explore: (1) knowledge graph completion with analogies, (2) transfer learning and zero-shot learning by analogy and (3) analogical question answering. We hope our work inspires future research on analogical reasoning and applications, especially in the multimodal world.

---

[4]A Huggingface Demo at `https://huggingface.co/spaces/zjunlp/MKG_Analogy`.

## REPRODUCIBILITY STATEMENT

The source MARS and MarKG datasets will be released on Github soon. In order to provide support to reproduce our experiments in Section 5, we provide the detailed source code of all pipeline baselines (IKRL, TransAE, RSME) and end-to-end baselines (VisualBERT, ViLBERT, ViLT, FLAVA, MKGformer) in the supplementary materials with all scripts and hyper-parameters. We also provide a README script to instruct how to run the codes.

## ACKNOWLEDGMENT

We would like to express gratitude to the anonymous reviewers for their kind comments. This work was supported by the National Natural Science Foundation of China (No.62206246 and U19B2027), Zhejiang Provincial Natural Science Foundation of China (No. LGG22F030011), Ningbo Natural Science Foundation (2021J190), and Yongjiang Talent Introduction Programme (2021A-156-G), CAAI-Huawei MindSpore Open Fund, and NUS-NCS Joint Laboratory (A-0008542-00-00).

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

## A  LIMITATIONS

The proposed work still has some limitations. We try to simulate the real-world multimodal analogy reasoning setting; however, it still can not predict analogical entities that are **not existing** in the multimodal knowledge graph. Such an issue is also known as inductive knowledge graph completion,

and we leave this for future works. Besides, we have not evaluated the very large-scale pre-trained models on the MARS due to the GPU resources, and it is well worth investigating whether large-scale pre-trained models can emerge the multimodal analogy reasoning ability.

# B ADDITIONAL DATASETS INFORMATION

## B.1 DATASET CONSTRUCTION

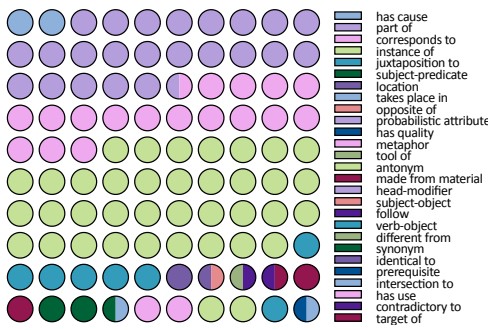

| | |
|---|---|
| has cause | |
| part of | |
| corresponds to | |
| instance of | |
| juxtaposition to | |
| subject-predicate | |
| location | |
| takes place in | |
| opposite of | |
| probabilistic attribute | |
| has quality | |
| metaphor | |
| tool of | |
| antonym | |
| made from material | |
| head-modifier | |
| subject-object | |
| follow | |
| verb-object | |
| different from | |
| synonym | |
| identical to | |
| prerequisite | |
| intersection to | |
| has use | |
| contradictory to | |
| target of | |

Figure 7: Relation distribution of MARS.

**Step 1: Collect Analogy Entities and Relations.** Since E-KAR and BATs are widely used text analogy datasets with high-quality and semantically specific entities, we collect the analogy seed entities $\mathcal{E}_a$ and relations from them according to the following criteria: (1) The relations and entities that have the same meanings will be merged. For example, we merge the relation *is_a* of E-KAR and the relation *Hypernyms* of BATs since they both represent the hypernym relationship of entities. We obtain 38 relations after this step. (2) The relation must imply analogical knowledge reasoning rather than simple word linear analogy. For example, we discard the analogy relations that only reflect simple word changes of BATs dataset such as *Inflections (Nouns, Verbs, etc.)* and *Derivation (Stem change, etc.)*. After this step, we filter 11 relations and retain 27 analogy relations.

(3) The entity must be visualizable and realistic. We filter those entities that cannot be linked into Wikidata and drop out the extremely abstract entities such as *virtue* by hand (some entities that have no image after Step 3 are also filtered). We discard a total of 463 entities after filtering. Finally, we obtained 2,063 seed entities and 27 relations.

**Step 2: Link to Wikidata and Retrieve Neighbors.** Consider that complex analogical reasoning is difficult through individual information (descriptions or images) of entities. We link the analogy seed entities to Wikidata by Mediawiki API [5] and retrieve the one-hop neighbors of seed entities as well as the possible relationships between the seed entities to obtain their neighbor structure information. In this step, we also take the retrieved descriptions from Wikidata as the textual information of entities and relations.

**Step 3: Acquire and Validate Images.** We collect images from two sources: Google Engine and Laion-5B query service[6]. We search from Google Engine with the descriptions of entities and crawl 5 images per entity. Laion-5B service depends on Clip retrieval and query by $k$nn index; we leverage the clip text embedding of the description and also query 5 images for each entity. Then we apply four filters to the above images: (1) we check the format of the images and filter invalid files, (2) we remove corrupted (the images are damaged and cannot be opened), low-quality (image size less than $50 \times 50$ or non-panchromatic images) and duplicate images, (3) we use CLIP (Radford et al., 2021) to remove the images with outlier visual embeddings, (4) we delete unreasonable images manually.

**Step 4: Sample Analogical Reasoning Data.** From Step 1 to Step 3, we obtain the MarKG, which includes 2,063 analogy entities, 8,881 neighbor entities, 27 analogy relations and 165 other relations. To construct the MARS dataset, we sample analogy example $(e_h, r, e_t)$ and analogy question-answer pair $(e_q, r, e_a)$ with the same relation $r$ from 2,063 analogy entities, but we do not explicitly provide

---

[5] https://www.wikidata.org/w/api.php
[6] https://knn5.laion.ai/

the relation in the input. Then we split the data into different task settings evenly. More details about the sample strategy of MARS can be seen in Section B.2.

## B.2 SAMPLE STRATEGY OF MARS

In Section B.1, we obtain the analogy seed entities $\mathcal{E}_a$ and the analogy relations between $\mathcal{E}_a$. Then we sample analogy example $(e_h, e_t)$ and analogy question-answer pair $(e_q, e_a)$ from $\mathcal{E}_a$. Guided by SMT, we make sure that $(e_h, e_t)$ and $(e_q, e_a)$ have the same relation $r$. Specifically, we divide the entity pairs that share the same relation into two categories to avoid overlap issues. Then we randomly sample the analogy examples from one category and the analogy question-answer pairs from another to construct analogy input instances. Last, we split the instances into different task settings evenly.

## B.3 DATASET DETAILS

|  | # entity | # relation | # triple | # image | source |
|---|---|---|---|---|---|
| WN9-IMG | 6,555 | 9 | 14,319 | 65,550 | WordNet |
| FB15k-IMG | 11,757 | 1,231 | 350,293 | 107,570 | Freebase |
| MarKG | 11,292 | 192 | 34,420 | 76,424 | Wikidata |

Table 5: Data statistics of MarKG. # refers to the number of.

The statistical comparison of MarKG with two multimodal knowledge graph datasets WN9-IMG (Xie et al., 2017) and FB15k-IMG (Liu et al., 2019) as shown in Table 5, we report the number of entity, relation, triple, image and the data source. Note that WN9-IMG and FB15k-IMG aim for knowledge completion and triple classification tasks while our MarKG aims to support MARS to do multimodal analogical reasoning. We also show the complete relations of our MARS in Table 6 and the distribution of relation categories in Figure 7.

| Relations | Definition | Example |
|---|---|---|
| part of | Object of which the subject is a part. | mouse : computer |
| corresponds to | Terms generally correspond to each other. | entrepreneur : laborer |
| juxtaposition to | Two terms belong to the same hypernym or have the same properties or functions. | child : minor |
| synonym | Sense of another lexeme with the same meaning as this sense. | tired : exhausted |
| made from material | Material the subject or the object is made of or derived from. | building : cement |
| antonym | Sense of a lexeme with the opposite meaning to this sense. | warm : cool |
| has cause | Underlying cause, thing that ultimately resulted in this effect. | cleaning : tidy |
| opposite of | Item that is the opposite of this item. | black : white |
| follow | The terms have a chronological or other sequential relationship, but one term does not cause the other. | implement : evaluate |
| intersection to | The extension of the two terms intersects. | odd : integer |
| takes place in | A term takes place in the other. | doctor : hospital |
| prerequisite | Prior event or achievement that a person or team needs to complete before joining or obtaining the item topic. | aim : shoot |
| subject-object | The originator and receiver of an action. | school : education |
| contradictory to | Two term are contradictory to each other. | english : chinese |
| identical to | The meanings of two terms are identical. | highway : road |
| head-modifier | The preceding term modifies the other. | affluence : living |
| different from | Item that is different from another item, with which it may be confused. | apple : nuts |
| probabilistic attribute | One term is probably the attribute of the other. | liquid : fluidity |
| instance of | That class of which this subject is a particular example and member. | coffee : drink |
| has use | Main use of the subject. | ballot : election |
| location | Location of the object, structure or event. | student : classroom |
| verb-object | The action and the object on which the action acts. | drilling : petroleum |
| has quality | The entity has an inherent or distinguishing non-material characteristic. | knife : sharp |
| tool of | One term is the tool of the other. | piano : play |
| subject-predicate | The originator of the action and the action itself. | stone : throwing |
| target of | One term is the target of the other. | harvest : sow |
| metaphor | A term is the metaphor of the other, reflecting something abstract indirectly. | pigeon : peace |

Table 6: The complete relations with definitions, examples of MARS. Some relations and definitions refer to (Chen et al., 2022a) and Wikidata Properties.

**Quality Control of Datasets.** We devise some quality control strategies while construct our MarKG and MARS datasets: (1) Entity and relation formalization and normalization. We link the analogy entities collected from E-KAR and SAT to Wikidata and filter non-link items. Since Wikidata is a knowledge base with quality-assured, some rare or worthless entities are excluded. (2) Image validation mechanism. We devise complex image filter strategies to control the robustness of image data, as mentioned in Section B.1. (3) Control of text description. We take the description in Wikidata as the textual information of entities.

| Method | Hit@1 | Hit@3 | Hit@5 | Accuracy |
|---|---|---|---|---|
| TransAE | 0.15 | 0.37 | 0.50 | - |
| MarT_VisualBERT | 0.15 | 0.28 | 0.53 | - |
| MarT_MKGformer | 0.16 | 0.36 | 0.59 | - |
| Human | - | - | - | 0.64 |

Table 7: Human evaluation on MARS.

**Human evaluation on MARS.** To evaluate the complexity and difficulty of the multimodal analogical reasoning task, we build a human evaluation in this section. However, humans encounter the following problems in this entity prediction task: (1) The candidate entity set is too huge for humans to select one entity. (2) Hit@k metric is not available since human hard rank predictions. Therefore, we utilize the multiple-choice format for human beings and apply the Accuracy metric to evaluate. Specifically, we randomly sample 100 instances from the test set to construct the evaluation set, and we use the top 10 ranking entities in TransAE prediction as candidate choices for each instance. If the golden answer entity is not in the top 10 entities, we will randomly replace one candidate with the golden entity. Then humans must select one entity from the candidate choices as the answer entity. The results can be seen in Table 7. We limit the prediction space of baseline models in candidate choices for a fair comparison. We find that the performance of the baselines in the Hit@1 metric has a large gap with human, which indicates the difficulty of the multimodal analogical reasoning task.

## B.4 DETAILED EVALUATION METRICS

The evaluation method of (Chen et al., 2022a) can not reflect one-to-more entities and does not fully explore the internal knowledge in the models due to the limited search space. Thus, we follow the link prediction task and choose Hits@k and MRR as our evaluation metrics. Both metrics are in the range $[0, 1]$. The bigger, the better performance. The Hits at k metric (Hits@k) is acquired by counting the number of times the golden entity appears at the first k positions in the predictions.

Given the prediction score of each entity in the candidate entity set, we sort the score and obtain the ranking of each entity. Denote the rank of the gold entity of $i$ triple as $rank_i$, and the reciprocal rank is $1/rank_i$. The Mean Reciprocal Rank (MRR) is the average of the reciprocal ranks across all triples in the knowledge graph:

$$\text{MRR} = \frac{1}{|\mathcal{S}|} \sum_{i}^{|\mathcal{S}|} \frac{1}{rank_i} \tag{8}$$

where $|\mathcal{S}|$ is the total number of the training set.

## C ADDITIONAL OF EXPERIMENTS

### C.1 IMPLEMENTATION DETAILS

This section detail the training procedures and hyper-parameters for various models. For multimodal knowledge representation methods, we first use MarKG to do knowledge representation learning and obtain the entity and relation matrix embeddings. Then we apply abduction and induction processes to continue training the models on the MARS dataset. Note that these processes are serial and share models. For multimodal pre-trained Transformer models, we also use MarKG to pre-train the models and then fine-tune on MARS end-to-end with our analogy prompt tuning strategy. We utilize Pytorch to conduct all experiments with 1 Nvidia 3090 GPU. The details of hyper-parameters can be seen in Table 8.

| Hyper-parameters | MKGE Baselines | MPT Baselines |
|---|---|---|
| epoch | {300, 1000} | 15 |
| sequence length | - | 128 |
| learning rate | {1e-2, 5e-3} | {3e-5, 4e-5, 5e-5} |
| batch size | 1000 | 64 |
| optimizer | {Adagrad, SGD} | AdamW |
| adam epsilon | - | 1e-8 |
| $\lambda$ | - | {0.38, 0.43, 0.45} |

Table 8: Hyper-parameter settings. We use the same parameter settings of MKGE baseline methods as the original paper except for the learning rate.

## C.2 RESULTS OF PRE-TRAINING ON MARKG.

| Method | Baselines | Entity Prediction | | | | Relation Prediction | | | |
|---|---|---|---|---|---|---|---|---|---|
| | | Hits@1 | Hits@3 | Hits@10 | MRR | Hits@1 | Hits@3 | Hits@10 | MRR |
| MKGE | IKRL | 0.157 | 0.257 | 0.338 | 0.272 | - | - | - | - |
| | TransAE | 0.307 | 0.361 | 0.442 | 0.353 | - | - | - | - |
| | RSME | 0.417 | 0.460 | 0.520 | 0.452 | - | - | - | - |
| MPT | MarT_VisualBERT | 0.466 | 0.598 | 0.692 | 0.546 | 0.758 | 0.873 | 0.927 | 0.822 |
| | MarT_ViLT | 0.466 | 0.586 | 0.675 | 0.539 | 0.737 | 0.847 | 0.902 | 0.799 |
| | MarT_ViLBERT | 0.489 | 0.621 | 0.711 | 0.569 | 0.764 | 0.876 | **0.930** | 0.827 |
| | MarT_FLAVA | 0.506 | 0.634 | 0.716 | 0.582 | **0.771** | **0.877** | 0.921 | **0.829** |
| | MarT_MKGformer | **0.527** | **0.670** | **0.779** | **0.616** | 0.762 | 0.870 | 0.923 | 0.823 |

Table 9: Pre-training results on MarKG. Note that these results are from the training process as we do not divide MarKG. Since we follow the link prediction task to pre-train the models for MKGE baselines, we only report the entity prediction results.

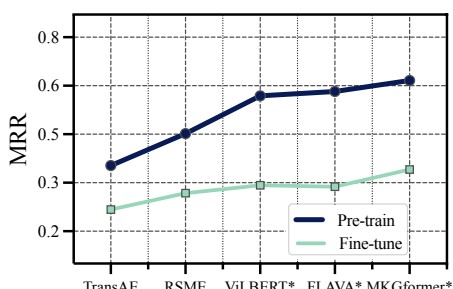

Figure 8: The results of pre-training on MarKG and fine-tuning on MARS. * refers to the baseline model applied MarT.

We report the pre-train results on MarKG in Table 9. We find that MPT baselines perform better than MKGE baselines consistently. It reveals the strong fit ability of Transformer-based models. As shown in Figure 8, we can observe that pre-training and fine-tuning stages trends are roughly the same, especially in the same type of baselines, which indicates that pre-train on MarKG benefits analogical reasoning on MARS.

## C.3 RESULTS OF IMPLICIT RELATION INFERENCE OF MPT.

We conduct an evaluation experiment on the relation inference of MKGE and MPT methods. For MKGE methods, we evaluate the relation predicted of *Abuduction* process with hit@k metrics. Since MPT methods solve the analogical reasoning task end-to-end without any explicit relation prediction process, we use two ways to evaluate their relation-aware abilities. The first one is that we predict the relation via the special relation token [R], which is similar to mask entities prediction and evaluate the predictions with Hit@k metrics. However, this evaluation method does not precisely reflect the relation-aware abilities of models since [R] is an abstract virtual token that may aggregate multiple

| Method | Baselines | Relation Prediction | | | Distance |
|---|---|---|---|---|---|
| | | Hits@3 | Hits@5 | Hits@10 | |
| MKGE | IKRL | 0.160 | 0.234 | 0.405 | - |
| | TransAE | **0.179** | 0.242 | 0.491 | - |
| MPT | MarT_VisualBERT | 0.107 | 0.181 | 0.340 | 1.418 |
| | MarT_ViLT | 0.126 | 0.181 | 0.332 | 1.419 |
| | MarT_ViLBERT | 0.078 | 0.189 | 0.333 | 1.412 |
| | MarT_FLAVA | 0.078 | **0.587** | **0.709** | **1.380** |
| | MarT_MKGformer | 0.049 | 0.209 | 0.512 | 1.405 |

Table 10: Relation evaluation of MPT baselines.

relation information. Therefore, we devise the second method that computes the Euclidean distance as follows:

$$\text{Distance} = \frac{1}{|\mathcal{S}_t|} \sum_i^{|\mathcal{S}_t|} d(\text{Norm}(h_{[\text{R}]}^{\mathcal{A}}), \text{Norm}(E_{e_r})) \quad (9)$$

where $|\mathcal{S}_t|$ is the total number of the test set, $h_{[\text{R}]}^{\mathcal{A}}$ is the hidden state of [R] in the last transformer layer, $E_{e_r}$ is the special relation embedding (described in Section 4.2.1) of the golden relation $r$. $\text{Norm}(\cdot)$ is the $l2$-normalization of vectors, $d(\cdot, \cdot)$ is the Euclidean distance function.

The evaluation results are shown in Table 6, we find that MKGE methods perform better than most MPT methods on Hit@k metrics, especially on Hit@3 metric, which may benefit from the explicit relation perception in the pipeline process. Moreover, MarT_FLAVA achieves the best relation-aware performance on Hit@k and Euclidean distance metrics, but MarT_FLAVA performs worse than MarT_MKGformer in answer entity prediction as shown in Table 2. We speculate that the special token [R] contains not only the golden relation but also other related relation information.

## C.4 COMPARISON OF PERFORMANCE AND MODEL SIZE

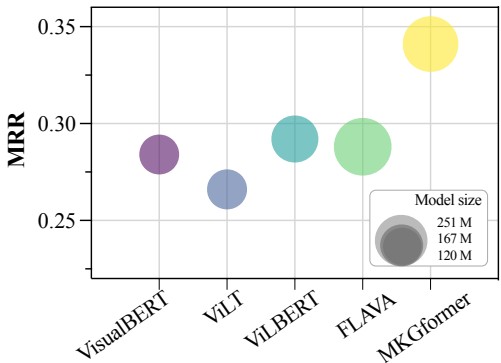

Figure 9: Comparison of performance and model size of MPT baselines.

In this section, we detail the size of MPT baseline models and compare them with their performance. In MPT models, the single-stream models (VisualBERT, ViLT) are the smallest, the dual-stream models (ViLBERT) are the middle and the mixed-stream models (FLAVA, MKGformer) are the biggest. The performance of models is roughly proportional to their sizes, as shown in Figure 9. MKGformer outperforms all other models, including the biggest FLAVA model.

## D ERROR CASE ANALYSIS

In this section, we conduct an error case study on MARS in Figure 10. From the error cases, we can see the hardship of the multimodal analogical reasoning task: 1) **Imbalance of multimodal.** The semantic scales of images and text are inconsistent, which leads to incorrect matching (Zhu et al., 2022). Although we filter some hard-to-visualize entities in data collection in Section B.1,

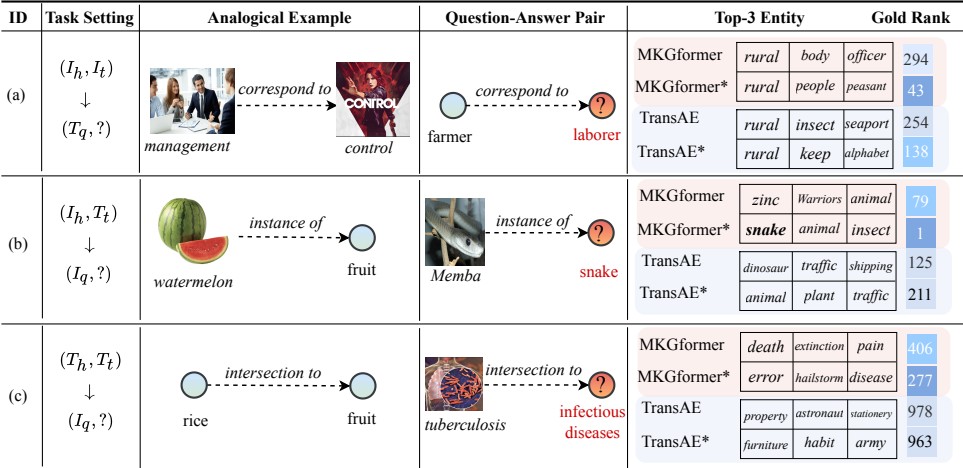

Figure 10: Error case examples.

the high semantic entities exist. As shown in example (a), "management" and "control" are abstract entities that are difficult to find equivalent images. Moreover, the uncoordinated convergence problem in multimodal learning further exacerbates the difficulty of the multimodal analogical reasoning task (Peng et al., 2022; Wang et al., 2020). 2) **One-to-more problem.** It is challenging for the models to solve one-to-more entities. In example (b), "Memba" is an instance of both "snake" and "animal", which is confusing to MKGformer. 3) **Unintuitive relations.** In our MARS dataset, some relations are not intuitive, requiring models to have strong relation reasoning ability. As shown in example (c), the relation "intersection to" means the extension of the head and tail entity intersects. All four models are struggling and far away from the golden answer entity.

