# OpenReview forum: "Multimodal Analogical Reasoning over Knowledge Graphs"
_ICLR.cc/2023/Conference — ICLR 2023 poster_

### Official Review · Reviewer_R5x6 · 2022-10-24

**Confidence:** 5
**Clarity, Quality, Novelty And Reproducibility:** see above
**Correctness:** 3
**Technical Novelty And Significance:** 2
**Empirical Novelty And Significance:** 3
**Recommendation:** 5

**Strength And Weaknesses:**

Strengths:
1. The motivation is reasonable and the paper is mostly clear.
2. The paper contributes a new task and dataset for multimodal reasoning.

Weaknesses:

1. The presentation needs further improvements. For example, 1) fig. 3 and Section 4.2.1 is not clear. What exactly are the e_h and e_t in the inputs? Why are there two [PAD]? Are there two inputs or just one? What is the [MASK] for? why the [MASK] needs to multiply E_e1 and E_ri? 2) How to ensure the quality of the proposed dataset? Even though I have checked the Appendix, I still cannot find the annotation guideline or agreement among annotators.

2. The experiments need further discussion to show the necessity of the task and the quality of the dataset. 1) The multi-modal analogical reasoning task can be decomposed into two sub tasks: cross-modal alignment and single-modal analogical reasoning. What is the advantage of evaluating them at the same time? 2) What is the main reason for the unsatisfactory performance, low-quality data? difficult cross-modal alignment or analogical reasoning? I would suggest to evaluate the two sub-tasks separately for further investigation. 3) From the cases in fig. 6, the predicted entities varies a lot. Why?

**Summary Of The Paper:**

The paper proposes a new task of multimodal analogical reasoning that takes the inputs of a pair of head and tail entities in different modalities, and predicts the missing tail entity for a given query head entity. The task is inspired by conventional single modal analogical reasoning and cognitive theory: human learns well from multiple modal sources instead of a single one. To build the benchmark, the authors collect images from Google for textual entities in KG, and conduct several pre-processing steps to clean the data. Furthermore, three baseline methods are proposed, including MKGE methods and pre-training methods.


**Summary Of The Review:**

see above

---

> ### Author Response · Authors · 2022-11-16
> **Response to Reviewer R5x6 (Part 1)**
>
> Dear reviewer R5x6,
>
> Many thanks for your detailed and constructive comments.
>
> Response to Weakness 1:
>
> > Q. What exactly are the e_h and e_t in the inputs?
>
> Sorry for the unclear parts. We represent each entity and relation as a special token and add them to the word vocabulary of transformer models. The e_h, e_t and e_q are entity ids in the word vocabulary and will be encoded to special entity tokens in the word embedding layer. We add e_h, e_t and e_q in the input because their special embedding can encode the corresponding entities' information.
>
> > Q. Why are there two [PAD]? Are there two inputs or just one?
>
> For all multimodal transformer-based methods, their inputs contain a visual part and a textual part. If the input does not include visual images, we input blank images and mask them in transformer layers. The number of the [PAD] is irrelevant since they are blinded for models. The two [PAD] are just for alignment with others.
>
> > Q. What is the [MASK] for? why the [MASK] needs to multiply E_e1 and E_ri?
>
> The role of the [MASK] token in our paper is similar to Masked Language Modeling (MLM) task. In the MLM task, some tokens of the input sentence are masked, and the final hidden vector corresponding to the mask tokens are obtained to make prediction over the word embeddings (multiply the word embedding matrix). The masked entity and relation prediction task are similar, except that the word embeddings are replaced by the entity and relation embeddings (E_e1 ..., E_ri, ...). This approach is also applied by Choi et al.[1] and Chen et al.[2] with remarkable results.
>
> > Q. How to ensure the quality of the proposed dataset?
>
> The quality of the proposed dataset can be ensured as follows: (1) The analogy entities and relations are from E-KAR and BATs datasets and then normalized by Wikidata. Since the E-KAR and BATs are two well-constructed analogical reasoning datasets in NLP, the quality of the analogy data can be ensured. (2) We devise many filter strategies, including entity and relation filtering and image filtering. Moreover, we conduct manual filters in the filter steps. (3) We conduct a human evaluation experiment in the revised draft. There are no annotation guidelines or agreement among annotators since only some manual filtering steps in our data construction process do not involve the annotation step. We detailed the construction steps in Appendix B.1. Sorry for the confusion.

---

> > ### Author Response · Authors · 2022-11-16
> > **Response to Reviewer R5x6 (Part 2)**
> >
> > Response to Weakness 2:
> >
> > > Q. What is the advantage of evaluating cross-modal alignment and single-modal analogical reasoning at the same time?
> >
> > The final practical effect is similar to direct reasoning with unimodal information if we first evaluate cross-modal alignment and then single-modal analogical reasoning.
> >
> > While our motivation for the multimodal setting is from Mayer's Cognitive Theory[3] that humans perform better on recall and transfer tests when they have learned from multimodal sources than from single-modal sources.
> >
> > We want to explore whether it is established for neural models. However, if we align the different modalities in advance, it cannot confirm our motivation. Thus, we treat the visual and textual modalities equally, and the models need to make analogies from multimodal information. The cross-modal alignment occurs implicitly within the model rather than occurs before input.
> >
> > > Q. What is the main reason for the unsatisfactory performance, low-quality data? difficult cross-modal alignment or analogical reasoning?
> > The reasons for the unsatisfactory performance can be regarded as follows:
> >
> > - The background knowledge challenges. Since most analogy entities are specific, the model needs to understand the meaning of these entities in order to make correct reasonings. Although the knowledge graph dataset MarKG provides some information about the entities, it is still incomplete. Moreover, we conducted a human evaluation experiment in Appendix B.3. Even though humans are familiar with most entities, the correct rate is only 0.64, which reveals the difficulty.
> >
> > - Most previous analogical reasoning researches are in the form of (a, b) : (c, d) and defined the task as a classification task. We set the reasoning form as (a, b) : (c, ?) and formulate the task as link entity prediction without explicitly providing relations. This task formulation is more composite to humans. The label space of our task is vastly expanded compared to the classification task, and the prediction difficulty is also tremendously increased.
> >
> > Furthermore, multimodal information is not the culprit for poor performance. As shown in Figure 5 (Section 5.4), the models perform better in the blended task setting than the single task setting, although the blended task setting contains more complex multimodal information.
> >
> > > Q. From the cases in fig. 6, the predicted entities varies a lot. Why?
> >
> > The analogical components can help models mine implicit relations and make more reasonable predictions. In particular, the relaxation loss allows the model to focus more on relation structure transfer rather than the similarity of entities when entities are complex. However, all models fail in some scenarios, as shown in Figure 9 (Appendix D).
> >
> > Thanks again for your time and precious advice!
> >
> > [1] MEM-KGC: Masked Entity Model for Knowledge Graph Completion With Pre-Trained Language Model. IEEE Access 2021.
> >
> > [2] Hybrid Transformer with Multi-level Fusion for Multimodal Knowledge Graph Completion. SIGIR 2022.
> >
> > [3] Multimedia learning. Psychology of learning and motivation 2002.

---

> > > ### Comment · Reviewer_R5x6 · 2022-11-26
> > > **Remain the score**
> > >
> > > Thanks for your response, which solves most of my concerns. However, the quality of the datasets is still unclear. First, the human evaluation should be in the content as the dataset is one of the main contributions in the paper. Second, the human evaluation experiment in appendix b.3 still lacks of guideline about what is a better choice and the description about annotators, such as agreements, the background of annotators, the number of annotators. I understand the difficulty in conducting human evaluation on the entire entity vocabularity, but this is very important for a dataset paper. Besides, the citation [1] seems irrelevant to the Mary's cognitive theory, I didn't find anything about multi-modal analogy reasoning.

---

> > > > ### Author Response · Authors · 2022-11-26
> > > > **Thanks for your feedback**
> > > >
> > > > Dear reviewer R5x6,
> > > >
> > > > Thank you so much for your feedback. We highly value each of your comments and concerns. We agree with you about the importance of human evaluation for a new dataset, and we will add the human evaluation experiment to the main text. For the quality of our datasets, we try our best to control it as follows:
> > > >
> > > > 1. The analogy entities and relations are from two well-constructed analogical reasoning NLP datasets E-KAR[1] and BATs[2], and are further linked into Wikidata by us.
> > > >
> > > > 2. The textual descriptions of the entities and relations are also from Wikidata, provided by the wiki page of entities. These descriptions are strictly aligned with entities and contain rich semantic information.
> > > >
> > > > 3. The visual images of the entities and relations are crawled from the Google engine and Laion-5B. We devise many filter strategies to ensure the relevance of images and entities.
> > > >
> > > > 4. We follow reviewer Hqif and conduct a human evaluation experiment.
> > > > There are no annotation guidelines or agreement among annotators since only some manual filtering steps in our data construction process do not involve the annotation step. We do not describe the evaluators in detail because common sense is enough to answer the questions in our dataset evaluation. But we will add more detailed descriptions and continue to improve our paper.
> > > >
> > > > We hope the above can address some of your concerns. Besides, the datasets are part but not all of our contribution. We sincerely hope to address your other concerns about the new multimodal analogical reasoning task, the framework MarT, the benchmark results of MKGE and MPT methods and the analysis.
> > > > Last, we are so sorry for the wrong citation, the Mayer's Cognitive Theory is citation [3]. We have corrected this mistake in [Response Part 2](https://openreview.net/forum?id=NRHajbzg8y0P&noteId=v6EZ8NdVqWa).
> > > >
> > > > **References**
> > > >
> > > > [1] E-KAR: A benchmark for rationalizing natural language analogical reasoning. ACL 2022.
> > > >
> > > > [2] Analogy-based detection of morphological and semantic relations with word embeddings: what works and what doesn't. NAACL 2016.
> > > >
> > > > [3] Multimedia learning. Psychology of learning and motivation 2002.

---

### Official Review · Reviewer_Hqif · 2022-10-26

**Confidence:** 4
**Correctness:** 3
**Technical Novelty And Significance:** 3
**Empirical Novelty And Significance:** 2
**Recommendation:** 5

**Clarity, Quality, Novelty And Reproducibility:**

The paper is not very well written in my opinion. There are various typos and it is sometimes hard to understand.

I still have some questions:
1. What are e_h, e_t, and e_q in (1)? Are they special tokens? This is not mentioned in the paper. 4.2.1 mentioned that all the entities and relations are special tokens, but for a question, we do not know the ground truth entity id for I_h and I_t.
2. Appendix B.1, Step 1: how do you perform the filtering of entities and relations? Did you check them manually? The criteria is vague.
3. B.1 Step 3: Did you get 5 images for all the 11k entities? Are all the entities visualizable? I wonder if some entities are quite abstract to represent with an image.

Typos:
1. Sec 4.3, adaptive Interaction: It should be "noisy data" instead of "noise data".
2. Sec 4.3, relation oriented: are "mapped" from, instead of are "mopped" from.

The paper proposes novel ideas for MAR. Codes and data are included with the submission.

**Strength And Weaknesses:**

Strength: The paper gives a novel KG along with the dataset for MAR. The authors also propose a method for MAR with transformers and show solid improvement over 5 different backbones.

Weakness:

1. The methods proposed for MAR are not very novel, as it is quite common among existing KG and language model methods.
2. Essentially, the proposed dataset is just an augmentation of E-KAR and BATs with images - the analogy data is already existing and the new things are just the images. If I understand correctly, MARS does not contain any data from Wikidata, and they do not have any tasks on MarKG. Therefore the wikidata part in MarKG is just for training purposes.
3. I think a human evaluation is essential for the MARS dataset. We need to know how human performs on this task to get an idea of the data quality.



**Summary Of The Paper:**

The paper proposes a new dataset and knowledge graph for multimodal analogical reasoning.
The knowledge graph (MarKG) is based on E-KAR and BATs, augmented by Wikidata and image search results from Google and Laion-5B. The dataset (MARS) is based on analogy relations in E-KAR and BATs; the task is that given two entities (e_h, e_t) with hidden relation r, and a question entity e_q, to predict the entity e_a with the same relation r to e_q. The given entities can be in an either textual or visual format.

The paper also proposes several methods for multimodal analogical reasoning (MAR). The first method uses existing multimodal entity embedding methods, with the ANALOGY loss from previous work. The second method uses a pretrained multimodal transformer (MPT). The authors first pretrain the MPT over MarKG by linearizing the KG; then they propose two modifications. The first one is multiplying the attention probability between head and tail entities by a learned parameter. The second is an additional loss that brings the contextual relation representation closer and entity representation further.

The results show that the proposed methods can improve the performance over naïve MPTs. MKGE methods perform competitively on this task, and only MKGformer can outperform the MKGE methods among the 5 MPTs that the authors tried.

**Summary Of The Review:**

In all, I feel the paper has good contributions, but the quality is preventing me from giving a higher score.

---

> ### Author Response · Authors · 2022-11-16
> **Response to Reviewer Hqif (Part 2)**
>
> **Response to Questions:**
>
> **For Q1.** Sorry for the unclear parts. We represent each entity and relation as a **special token** and add them to the word vocabulary of transformer models. The e_h, e_t and e_q are **entity ids** in the word vocabulary and will be encoded to special entity tokens in the word embedding layer. The embedding of these special tokens can encode the information of entities after the pre-training step (Section 4.2.1) and works in the masked prediction phrase. This approach is also applied by Choi et al.[1] and Chen et al.[2] with remarkable results. I_h and I_t are images of the entity e_h and e_t, and T_q is the textual description of entity e_q. These are provided in the input, so we actually know their ground truth entities.
>
> **For Q2.** We filter the entities and relations in the following steps:
> 1. We **merge** the entities and relations with the same meanings in E-KAR and BATs datasets. For example, the relation _is_a_ of E-KAR and the relation _Hypernyms_ of BATs will be merged since they both represent the hypernym relationship of entities.
> 2. We filter the **word linear analogy relations**. For example, we filter the relation _Inflections (Nouns, Verbs, etc.)_ and _Derivation (Stem change, etc.) _of BARTs because they only reflect simple word changes.
> 3. We filter those entities that **cannot be linked** into Wikidata and drop out the extremely **abstract entities**, such as _virtue_ by hand (some entities that cannot be retrieved from Laion-5B are also filtered).
>
> We discard a total of 463 entities and 11 relations after filtering. Step (1), step (2) and the abstract entities are all filtered manually and carefully checked.
>
> **For Q3.** We initially crawl 5 images per entity from Google Engine and query 5 images from Laion-5B, so we get 10 images per entity. After filtering, the number of images per entity is not fixed. Not all entities are visualizable and such entities will be filtered by two approaches: manually in Appendix B.1 step 1.(3) of the revised draft and drop out the entities with no image after image filtering.
>
> **For typos:** Thanks for your suggestions. We fixed these typos in the revised draft. But the word "mopped" is from Structure Mapping Theory[3]; We added some description around the sentence to clarify.
>
> **Other Response:**
>
> We are so sorry for the lack of detailed descriptions of our datasets and method. The dataset construction steps contain a lot of manual filters and are carefully checked. We have detailed the construction steps and confusion of methods in the revised draft. We also conducted a human evaluation following your suggestion in Appendix B.3. We sincerely hope our response can address some of your concerns.
>
> Thanks again for your time and precious advice!
>
> [1] MEM-KGC: Masked Entity Model for Knowledge Graph Completion With Pre-Trained Language Model. IEEE Access 2021.
>
> [2] Hybrid Transformer with Multi-level Fusion for Multimodal Knowledge Graph Completion. SIGIR 2022.
>
> [3] Structure-mapping: A theoretical framework for analogy. Cogn. Sci. 1983.

---

> ### Author Response · Authors · 2022-11-16
> **Response to Reviewer Hqif (Part 1)**
>
> Dear reviewer Hqif,
>
> Many thanks for your detailed and constructive comments.
>
> **Response to Weakness:**
>
> **For W1.** Note that the core contribution of our paper mainly includes the new multimodal analogical reasoning task over knowledge graphs, the new Multimodal Analogical Reasoning dataSet (**MARS**, construct based on E-KAE, BARTs, Wikidata and image sources) and a new knowledge graph named **MarKG** to support reasoning. We aim to call on researchers to focus on analogical reasoning problems that align with real scenarios.
> Another line of our contribution is designing a **pipeline** approach for MKGE and **end-to-end** method **MarT** for transformer-based models to make them adapt to multimodal analogical reasoning. Among them, MarT is a Multimodal analogical reasoning framework with Transformer, which contains adaptive association gates and a relaxation loss to improve. To the best of our knowledge, there are no similar operations in existing KG-related methods.
>
> **For W2.** We are so sorry for the lack of detailed data descriptions. We have conducted a supplement in Appendix B.1 Dataset Construction in the revised draft. Note that our proposed dataset is far from image augmentation of E-KAR and BATs.
> We just regard the entities and relations in E-KAR and BATs as the seed for linking rather than copying them with new images. Then we describe the detailed data construction as follows.
> In the collection of the analogy seed entities and relations step, we merge the entities and relations of E-KAR and BATs and filter simple word linear analogy relations (such as Nouns, Verbs, and Stem changes). Then we **link the entities to Wikidata** and filter unlinkable entities. Meanwhile, the descriptions of the entities and relations from Wikidata are collected as textual description information. The images of the entities are from Goole Engine and Laion-5B. So far, the entities and relations are **no longer** in the E-KAR and BATs but the entities and relations **formalized** **by Wikidata**. Moreover, the **textual descriptions** of the entities and relations are also from Wikidata. MarKG also has the neighbor information of the analogy entities retrieved from Wikidata, and the original intention of MarKG is to provide prior knowledge of analogy entities and relations for better reasoning.
>
> **For W3.** Thanks for your suggestion. We have conducted a human evaluation and detailed it in Dataset Details of Appendix B.3 in revised draft. However, there are two problems for human evaluation in entity prediction: (1) The candidate entity set is too huge for humans to select one answer entity. (2) Hit@k metric is not available since human hard rank predictions. To avoid the above issues, we devise an evaluation strategy to reduce candidate entities. Specifically, we randomly sample 100 instances from the test set as the evaluation set, and we use the top 10 ranking entities in TransAE prediction as the candidate choices (add the golden answer entity if it does not exist) for each instance. Humans must select one entity from the candidate choices as the answer, and we use the _Accuracy_ metric to evaluate. The results are as follows:
>
> | **Method**     | **Hit@1** | **Hit@3** | **Hit@5** | **Accuracy** |
> | --------------- | --------- | --------- | --------- | ------------ |
> | TransAE         | 0.15      | 0.37      | 0.50      | -            |
> | MarT_VisualBERT | 0.15      | 0.28      | 0.53      | -            |
> | MarT_MKGformer  | 0.16      | 0.36      | 0.59      | -            |
> | Human           | -         | -         | -         | 0.64         |
>
> We choose some baseline models to make a comparison. Note that we limit the prediction space of these models in the candidate choices for a fair comparison. We find that the performance of the baseline model in the Hit@1 metric has a large gap with humans, which indicates the difficulty of the multimodal analogical reasoning task. We hope this evaluation can help to understand dataset quality to some extent.

---

### Official Review · Reviewer_JzuQ · 2022-11-04

**Confidence:** 3
**Correctness:** 4
**Technical Novelty And Significance:** 3
**Empirical Novelty And Significance:** 3
**Recommendation:** 8

**Clarity, Quality, Novelty And Reproducibility:**

**Clarity:**
* While there are numerous grammar issues in the text, the text still flows well and is quite readable. this was **not** used against the authors in the review.
* Precisely what the inputs to the neural networks are is not extremely clear. Notation and descriptions could be improved to remove any doubt.
* Figure 4 is a bit hard to understand. Perhaps it's better to make it understandable standalone (i.e. it makes sense without referring to the text).
* Figure 3a: Why is the answer "Young Tesla", not just "Tesla"?

**Quality:** I believe that this submission is high quality, regarding the professionality of the writing/figures and the technical content.

**Novelty:** To the extent of my knowledge, the task, datasets, evaluations and the contents of the discussion are all sufficiently novel.

**Reproducibility:** While the paper could describe the model tuning procedure a bit more, all the results can likely be verified thanks to the dataset and code release. (I have not checked the code)

**Strength And Weaknesses:**

**STRENGHT:**
* **Strong motivation:** Given the increasing interest in multimodal reasoning, I believe that this paper is filling an important gap in literature. Just the dataset and task definition alone constitute strong contributions.
* **Multiple strong contributions:** The benchmarking and ablation experiments are comprehensive enough that they alone (i.e. excluding the dataset) could perhaps constitute a separate submission.
* **Dataset and test definition sensible:** Both the way the dataset is constructed (about which I'll list some clarifying questions below), the input and outputs are formatted and the way the models are evaluated (pass@k) seem quite sensible, and likely can be used without much modification by other researchers interested in multimodal analogical reasoning.

**WEAKNESSES:**
* **"No input pairs" baseline is missing:** It would be great if the authors evaluated the performance of the models when they are trained **without** with the analogy pair to condition on. That is, instead of following the current format, which is "if x is to y, then z is to ?', drop the first bit and only train and test on "z is to ?". This would give us the base rates and the performance obtained by this model would be very useful to make sense of how high the reported accuracies are.
* **Missing scaling plots:** While providing scaling plots is not standard practice today (which is why this critique should not be held strongly against the authors), I believe they provide priceless information that's hard to substitute. For example, I remain skeptical that the modification introduced in the self-attention equations (the learned gating) will continue to matter at larger model scales. Scaling plots can serve to dispel these suspicions.
* **No baseline without explicitly inferring relations:** While the choice to stick to the "abduct-relate-infer" pipeline is sensible, it still remains to be seen whether DL models can still perform well without this explicit structure. The submission would be stronger if the authors at least reported some preliminary results on the effect of relation inference.
* **(Nitpick) Multi-pair instances:** This is not really a weakness, but perhaps a sensible next step(?): It seems relatively easy to convert this dataset into a k-shot analogy reasoning one. That is, instead of a single analogy pair, there could be k number of them. Perhaps this could help authors reach new conclusions that cannot be reached using the current version of the dataset.
* **A bit confusing sentence:** What exactly do you mean by this: "The structure mapping theory described that relations
between objects, rather than attributes of objects, are mopped from base to target."
* **(nitpick)** Results in Table 2 don't **prove** Mayer's theory, but support it. Just wanted to highlight this!




**QUESTIONS TO AUTHORS:**
* **Entity filtering:** You mention in the Appendix that you constrain the entities to be visualizable. How did you enforce this? Also what does the following mean exactly? "the relation must imply analogical knowledge reasoning rather than simple word linear analogy"
* **Removing corrupted and low quality iamges:** How does this step, also outlined in appendix, work exactly?
* **MarKG vs. MARS:** Could you precisely describe what the connection between these is? How much of an overlap is there between these?
* **Text description vs. entity token:** This is perhaps a big misunderstanding, but I didn't quite get the distinction between what the difference between a "text description" and an "entity token" is.
* **prompt tuning:** What exactly do you mean by "prompt tuning" in Remark 1? Prompt tuning is a method to train the context embeddings by backpropping on them using a given task loss. I presume you don't mean this?
* **Gates in MarT:** If these gates are learnable parameters, how do they adaptively adjust how much cross-entity attention happens?
* **Position embeddings:** What position (if any) do you use? The gates in MarT resemble a more restricted version of T5-style position encodings to me. Is this right?
* **Noise data:** Just wanted to clarify whether you mean "noisy data" in Remark 2? In either case, I don' quite get this point.
* **Generalizing to novel relations:** 1) What about the MarKG is helping the models acquire this capability? 2) You have some ablations without MarKG pretraining that still obtain nontrivial accuracy. How is this possible, given that the model has never interacted with a novel relation yet (even the embeddings for that relation is, as far as I understand, won't be learned)? More on this would be great!
* **Hyperparameter tuning strategy:** Could you outline the hyperparameter tuning strategies you used?


**Summary Of The Paper:**

**High level motivation:** While the multimodality of human perception is shown to help with analogical reasoning, there isn't a standardized task adopted by the machine learning community to drive progress in this area.

**Contributions:**
* **New task and dataset:** The authors propose a new multimodal analogical reasoning task over knowledge graphs. They construct a dataset named MARS (Multimodal Analogical Reasoning dataSet) to evaluate the performance in this task, and a knowledge graph named MarKG to support pretraining.
  * This contribution subsumes details regarding how the inputs/outputs should be formatted/prompted during pretraining and finetuning. Given that there are likely multiple ways to do this, this is a nontrivial contribution (I believe the authors' choices are sensible)
* **Extensive evaluation of existing benchmarks:** The authors run an extensive benchmarking evaluations and ablations to identify how the current SOTA approaches in analogical reasoning perform. They reach nontrivial conclusions, such as certain architectural components (especially the ANALOGY backbone by Liu et. al. (2017)).
* **Architectural and training modifications to boost performance:** The authors propose MarT, a framework that consists of a modification to the attention mechanism of a transformer and an added loss to support Structure Mapping from cognitive science. Their proposed modifications seem to lead to improvements in their ablations.
* **Insightful discussion:** The authors end the paper with an insightful discussion of the findings. Some highlights include the models' ability to generalize to novel relations (and hte effect of pretraining on this) and the role of the proposed relaxation loss.

**Summary Of The Review:**

This is a strong submission with clear contributions (unless there's omitted prior work that I'm not aware of).

I believe that the research community would be better off with this paper accepted to ICLR.

---

> ### Author Response · Authors · 2022-11-16
> **Response to Reviewer JzuQ (Part 2)**
>
> **Response to Questions:**
>
> **Q1. Entity filtering:** We first filter the **entities** that can not be linked into Wikidata and then drop out the extremely abstract entities such as "_virtue"_ by hand. Otherwise, the entity is also discarded if it have no image after image filtering in Appendix B.1 step 3. Finally, We discard 463 entities in total. Furthermore, some relations in the BATs dataset only reflect simple word changes (we call it _simple word linear analogy_), such as "_Inflections" (Nouns, Verbs, etc.)_ and "_Derivation" (Stem change, etc.)_. We discard these relations in the analogy relation collection step.
>
> **Q2. Removing corrupted and low quality images:** The corrupted images include damaged images that cannot be opened. The low-quality images include small-size images and non-panchromatic images. We remove the low-quality images during the crawling process and drop the corrupted images afterward.
>
> **Q3. MarKG vs. MARS:** MARS is a multimodal analogical reasoning dataset that includes analogy examples and question-answer pairs. The entity of MARS is called an analogy entity, and the relation is called an analogy relation. Moreover, the relation is implicit and not provided in the input.
> MarKG is a multimodal knowledge graph dataset in a (head, relation, tail) triplet format to support MARS. MarKG contains all analogy entities and relations but **not any triplet of MARS** (analogy examples and analogy question-answer pairs with implicit relations) to avoid label leakage. In addition, MarKG also has the neighbor information of the analogy entities retrieved from Wikidata. The original intention of MarKG is to provide prior knowledge of analogy entities and relations for better reasoning.
>
> **Q4. Text description vs. entity token:** We are sorry for the confusion. "entity token" is the special token we added to the transformer word vocabulary. It indicates a specific entity in MARS or MarKG. "entity token" is used to make **masked entity prediction**via its special token embeddings, similar to Masked Language Modeling (MLM) task. Text description and image are semantic information of the entity to help models learn the embedding of "entity token" and do prediction.
>
> **Q5. prompt tuning:** Prompt tuning in our methods mainly contains two areas: prompt template for input and masked entity prediction. The prompt template unifies the input and enables the models to make analogical reasoning end-to-end. The masked entity prediction reduces the gap between fine-tuning and pre-training of transformer-based models.
>
> **Q6 & Q7. Gates in MarT & Position embeddings:** The adaptive association gate is not motivated by the position encodings of T5. The position encodings of T5 are shared in all transformer layers, but our gates are learned in each transformer layer and perform different level associations.
>
> **Q8. Noise data:** Yes, it is "noisy data"; we have corrected this typo in the revised draft. There are some inevitable data in MARS that the analogy examples and analogy question-answer pairs have a weaker or uncorrelated connection. The adaptive association gates can alleviate this to a certain extent.
>
> **Q9. Generalizing to novel relations:** 1) The neighbor information (entities and relations) provided by MarKG is well structure prior knowledge. The model can learn rich representations of entities and relations (source and target), ), which benefit relation generalization. 2) We think there are two possible reasons to explain this. First is that the relations are not overlapped between source and target, but part entities are shared. The second point is that the end-to-end baselines without explicit relation structure are inclusive of relations. In these models, only the initialization stage of the special token [R] are affected by the novel relation embeddings, but they can aggregate multiple source relation information.
>
> **Q10. Hyperparameter tuning strategy:** For all experiments, we choose the model performing the best on the validation set and evaluate it on the test set. We apply grid search to find the best hyperparameters (batch size, learning rate, $\lambda$ in relaxation loss). Details can be seen in Appendix C.1.
>
> **Response to Clarity:**
>
> Thanks for your suggestion. We have revised the descriptions of the inputs and added some text annotation in Figure 4.a. The answer can be "Tesla" if the entity "Tesla" exists in the KG, but "Young Tesla" is more reasonable. This might be a one-to-more problem challenge for models (discussed in Appendix D Error Case Analysis). Our dataset has only a few one-to-more triplets (< 3%), and we take only one as the golden answer entity. We will extend our work on this issue in the future.
>
> Thanks again for your time and precious advice!

---

> > ### Comment · Reviewer_JzuQ · 2022-11-26
> > **Thank you for your response**
> >
> > I thank the authors for their detailed feedback.
> >
> > I still believe that this submission makes a number of valuable contributions that'd benefit the NeurIPSS community. Hence, I'm inclined to maintain my score for now, pending the reviewer discussion.
> >
> > Quick follow-up regarding Q9: You list two possible reasons that'd allow for knowledge transfer that'd enable generalizing to novel relations. Could you please elaborate on these a bit more, preferably with examples from the MarT and MarKG dataset (i.e. could you give one or few examples of a novel relation that, say, are supported by MarKG?)? I think this is a very interesting and nontrivial result of the paper, and further elaboration with specific examples and justifications would improve the paper. Since the paper update period has ended, I hope the authors will consider adding this in the camera-ready version should the paper be accepted.

---

> > > ### Author Response · Authors · 2022-11-27
> > > **Thanks for your feedback**
> > >
> > > Dear Reviewer JzuQ,
> > >
> > > Thank you so much for your feedback. We highly value each of your comments and concerns. Thank you very much for your recognition of our contributions. Then we explain the novel relation transfer experiment with an example:
> > >
> > > MarKG provides the prior structure information of entities and relations, similar to knowledge injection. Given an analogy example *(tiger, feline)* with the implicit relation *instance_of*, then ask the model what the answer of *(brain, ?)* is. Humans can easily get the gold answer *organ* because we understand each entity, naturally reasoning the *tiger* is an instance of *feline*, then we can analogy *instance_of* to *brain* and get the answer *organ*. However, neural models know little about the entities in the real world. Although models are trained on source data (source relation data in MARS) and awakened the sense of analogy, they still struggle with unfamiliar entities. Fortunately, MarKG tells models that *tiger* is a specie of big cat and habitat in forests; *feline* is cat-like animals and is the subclass of zoomorph; *brain* is also known as animal brain and is the organ that serves as the center of the nervous system; *instance_of* is a type constraints relation. (this information is stored as triples in MarKG.) With this prior information, models can make better predictions in the novel relation transfer experiment.
> > >
> > > We hope the above can address some of your concerns. We will add some examples in the form of figures or tables and conduct an in-depth analysis in the camera-ready version if our paper can be accepted.

---

> ### Author Response · Authors · 2022-11-16
> **Response to Reviewer JzuQ (Part 1)**
>
> Dear reviewer JzuQ,
>
> Many thanks for your detailed and constructive comments.
>
> **Response to weakness:**
>
> **W1. No input pairs baseline.** Thanks for your advice on the baseline. We have conducted experiments that input without the analogy example. The results can be seen as follows:
>
> | **Model**           | **Hits@1** | **Hits@3** | **Hits5** | **Hits10** | **MRR** |
> | ------------------- | ---------- | ---------- | --------- | ---------- | ------- |
> | MarT_ViLBERT        | 0.256      | 0.312      | 0.327     | 0.347      | 0.292   |
> | w/o Analogy Example | 0.113      | 0.143      | 0.162     | 0.179      | 0.138   |
> | Mart_MKGformer      | 0.301      | 0.367      | 0.380     | 0.408      | 0.341   |
> | w/o Analogy Example | 0.101      | 0.123      | 0.132     | 0.149      | 0.210   |
>
> The performance of models drops a lot when the analogy example is ablated. We hope this result can make sense of the main results. The revised draft has added more details in 5.3 (Ablation Study).
>
> **W2. Missing scaling plots:** Thanks for your suggestion. We added a bubble plot to compare the performance and size of MPT models in Appendix C.4 of the revised draft.
> However, we cannot experiment with larger MPT models since the sizes of their pre-trained weights are limited. (We download them at the huggingface website [2]) We compare the performance and model sizes of existing MPT models and find that the performance is roughly proportional to size.
>
> **W3. No baseline without explicitly inferring relations:** It's interesting to figure out the relation-aware abilities of the models without relation-explicit structure. We have conducted a relation prediction experiment on MKGE and MPT methods in Appendix C.3 of revised draft. For MKGE methods, we directly evaluate the relation predicted by the _Abduction_ process in hit@k metrics. For MPT methods, we predict the relation via the special relation token **[R],** which is similar to predict masked tokens and evaluate the predictions in Hit@k metrics. However, the special token **[R]** aggregates multiple relation information; this evaluation method is insufficient. Thus, we also compute the Euclidean distance between the hidden state of **[R]** ( $ h^A_{[R]} $ ) and the special relation embedding of the golden relation ( $ E_{e_r} $ ). The results are as follows:
>
> | **Method** | **Baselines**   | **Relation Prediction** |           |            | **Distance** |
> | ---------- | --------------- | ----------------------- | --------- | ---------- | ------------ |
> |            |                 | **Hits@3**              | **Hits5** | **Hits10** |              |
> | MKGE       | IKRL            | 0.160                   | 0.234     | 0.405      | -            |
> |            | TransAE         | **0.179**               | 0.242     | 0.491      | -            |
> | MPT        | MarT_VisualBERT | 0.107                   | 0.181     | 0.340      | 1.418        |
> |            | MarT_ViLT       | 0.126                   | 0.181     | 0.332      | 1.419        |
> |            | MarT_ViLBERT    | 0.078                   | 0.189     | 0.333      | 1.412        |
> |            | MarT_FLAVA      | 0.078                   | **0.587** | **0.709**  | **1.380**    |
> |            | MarT_MKGformer  | 0.049                   | 0.209     | 0.512      | 1.405        |
>
> Most MPT methods are inferior to MGKE methods, especially in the Hits@3 metric. Intuitively, the straightforward pipeline way facilitates the capture of the potential relation in the analogy example. However, the MPT methods' performance is inconsistent with the answer entity prediction; we think the implicit way can enrich the model with multiple relation information to do the analogy reasoning task.
>
> **W4. Multi-pair instances:** Thanks a lot for your excellent suggestion. Providing k-shot or non-fixed number analogy examples is a direction well worth exploring that is more in line with real-world human reasoning. We will follow this way to improve and enrich our dataset in the future.
>
> **W5. A bit confusing sentence:** We are sorry for the confusing sentence. Actually, the sentence comes from Structure Mapping Theory [1]; we cite it to emphasize the relation structure transfer is the crucial point in analogical reasoning rather than the object similarity between the source and target domain. In the revised draft, we have added some explanatory statements to eliminate confusion in Section 4.3 relaxation loss.
>
> **W6:** Thanks for your suggestion! We have corrected it in the revised draft.
>
> [1] Structure-mapping: A theoretical framework for analogy. Cogn. Sci. 1983.
>
> [2] https://huggingface.co/

---

### Author Response · Authors · 2022-11-16
**Summary of Revisions**

Dear Reviewers and AC:

We sincerely appreciate your valuable time and constructive comments.

We've uploaded a revised draft incorporating reviewer feedback. Modified text is shown in blue font. Below is a summary of the main changes:

- Section 4.2.4 states that the input of transformer-based modes and the prediction process.
- Section 4.3 optimizes Figure 4 and states that the relation transfer rather than object similarity in analogical reasoning.
- Add the ablation study that training and testing models without analogy examples in Section 5.3.
- Add more details about the dataset construction steps in Appendix B.1.
- Add human evaluation experiment on MARS in Appendix B.3.
- Add the results of implicit relation inference of MPT methods in Appendix C.3.
- Add the comparison of performance and model size in Appendix C.4.
- Correct the typos mentioned by reviewers.

We briefly introduce the motivation, contribution, benchmark methods and quality of datasets as follows:

Motivation: Analogical reasoning holds an important place in human cognition; it is interesting to explore multimodal reasoning for neural models with knowledge graphs.

Contribution:
- Advance the traditional setting of analogy learning by a new multimodal analogical reasoning task over knowledge graphs.
- Collect and build a dataset MARS with a multimodal knowledge graph MarKG.
- Report the performance of various multimodal knowledge graph embedding methods and multimodal pre-trained Transformer baselines.
- Further propose a framework MarT to enhance transformer-based methods.

Benchmark Methods:
- A explicit pipeline approach with Abduction-Mapping-Induction steps for MKGE methods.
- A end-to-end approach with prompt-tuning for MPT methods.
- A multimodal analogical reasoning framework MarT for transformer-based methods.

Quality of Datasets:
- Our datasets MARS and MarKG are built based on E-KAR and BATs, which are two well-constructed analogical reasoning datasets in NLP. Our datasets are also normalized by Wikidata.
- We devise many filtering strategies, including manual filtering.
- We conduct a human evaluation experiment on MARS in revised draft.

We sincerely hope our responses and revisions address all reviewers’ concerns.

We sincerely believe that these updates may help us better deliver the benefits of the proposed work to the ICLR community.
Thank you very much,

Authors.

---

### Author Response · Authors · 2022-12-04
**Follow-up response to all reviewers**

Dear reviewers, we sincerely appreciate any suggestions and welcome any more questions about our response.

All Authors.

---

### Decision · Program_Chairs · 2023-01-20

**Decision:**

Accept: poster

**Justification For Why Not Higher Score:**

Because as pointed by reviewers Hqif and R5x6, the existing dataset is created in a rather simple way which has some issues.

**Justification For Why Not Lower Score:**

Because this paper is the first to create a dataset for Multimodal Analogical Reasoning.

**Metareview: Summary, Strengths And Weaknesses:**

This paper argues for importance of multimodal analogical reasoning, introduces the new task of multimodal analogical reasoning over a knowledge graph. In particular, they construct a Multimodal Analogical Reasoning dataSet (MARS) and a multimodal knowledge graph MarKG. This paper is asking an important question about the role of multimodality in analogical reasoning and goes all the way to coming up with a task to answer that question which requires constructing a dataset. While the construction might seem as combining existing datasets, the problem framing and coming up with a way to build such a task is interesting and useful for the community. Therefore, I recommend accepting the paper.

**Note From Pc:**

if the above contains the word "oral" or "spotlight" please see: "oral" presentation means -> notable-top-5% and "spotlight" means -> notable-top-25%. As stated in our emails, we are disassociating presentation type from AC recommendations

**Summary Of Ac-Reviewer Meeting:**

The main contributions of the paper, its strengths/weaknesses were discussed.

JzuQ: Authors did a reasonable job in framing the multimodal analogy task to run experiments on image domain and ask analogy questions. Modern transformers are able to get reasonable performance from image tasks.

Hqif: The main shortcoming is that they only map existing dataset to wikidata and finds images. They start from text so the knowledge is from text.

R5x6: My main concern is the dataset quality, which is the main contribution. The author response does not  resolve my concerns such as the annotation background and agreement. The annotation is just filtering which weakens the contribution (also pointed by Reviewer Hqif)

During the meeting, it became more clear that JzuQ has more expertise in this area and is strongly supporting the paper. Hqif and R5x6 were scoring the paper below acceptance but did not have a strong opinion about it.